# Helper-based Adversarial Training: Reducing Excessive Margin to Achieve a Better Accuracy vs. Robustness Trade-off

**Rahul Rade** [1]   **Seyed-Mohsen Moosavi-Dezfooli** [2]

## Abstract

While adversarial training has become the de facto approach for training robust classifiers, it leads to a drop in accuracy. This has led to prior works postulating that accuracy is inherently at odds with robustness. Yet, the phenomenon remains inexplicable. In this paper, we closely examine the changes induced in the decision boundary of a deep network during adversarial training. We find that adversarial training leads to *unwarranted* increase in the margin along certain adversarial directions, thereby hurting accuracy. Motivated by this observation, we present a novel algorithm, called *Helper-based Adversarial Training (HAT)*, to reduce this effect by incorporating additional *wrongly* labelled examples during training. Our proposed method provides a notable improvement in accuracy without compromising robustness. It achieves a better trade-off between accuracy and robustness in comparison to existing defenses.

## 1. Introduction

It has been demonstrated that state-of-the-art deep neural networks (DNNs) are susceptible to intentionally crafted, human-imperceptible perturbations of the input (Szegedy et al., 2014; Goodfellow et al., 2015; Moosavi-Dezfooli et al., 2016). While several defense mechanisms (Papernot et al., 2016; Madry et al., 2018; Tramèr et al., 2018; Moosavi-Dezfooli et al., 2019) have been proposed to circumvent this issue, adversarial training (AT) (Madry et al., 2018) stands out as one of the most popular and effective methods to learn robust models. Nevertheless, it is known to negatively affect the accuracy on clean samples, thus leading to the much-debated trade-off between accuracy and robustness (Tsipras et al., 2019; Yang et al., 2020).

Consequently, in an endeavour to alleviate the accuracy vs. robustness trade-off, several modifications have been proposed to AT (Zhang et al., 2019; Wang et al., 2020; Wong et al., 2020; Rice et al., 2020; Zhang et al., 2020) with few works (Schmidt et al., 2018; Alayrac et al., 2019; Carmon et al., 2019) even advocating the use of external data for achieving notable gains in robustness. However, as evident in the analysis by Gowal et al. (2021), recent algorithms like TRADES (Zhang et al., 2019) and MART (Wang et al., 2020) only achieve a $1\%$ gain in robustness to $\ell_\infty$-perturbations with norm $8/255$ on CIFAR-10 (Krizhevsky, 2009) while seeing $\sim 2\%$ drop in accuracy when compared to AT in both the cases: with and without additional data. Hence, the progress in terms of reducing the gap between accuracy and robustness has been rather limited.

In this work, we attempt to improve and demystify the aforementioned trade-off. To this end, we a closer look at the effect of training with adversarial examples on the geometry of decision boundary learnt by deep networks. First, we demonstrate that AT leads to an excessive increase in the margin along the adversarial directions of the input space computed for a standard trained network. We refer to these directions as *initial adversarial directions*, the reason for such terminology will become clear in the later sections. Second, we identify a direct connection between the excessive directional margin and the accompanied reduction in accuracy caused by AT. Finally, we propose a novel adversarial training scheme to reduce the directional margin and thus, achieve a better accuracy without losing robustness. We call this algorithm, *Helper-based Adversarial Training (HAT)*, the name derived from the fact that we incorporate additional training examples to *help* impede excessive directional robustness and hence, improve clean accuracy. We also provide an extensive analysis of HAT and compare it with state-of-the-art AT methods.

## 2. Preliminaries

Consider the input space $\mathcal{X} \subseteq \mathbb{R}^d$. Let $f_{\boldsymbol{\theta}} : \mathcal{X} \to \mathbb{R}^C$ represent a deep neural network classifier parameterized by $\boldsymbol{\theta}$, where $C$ is the number of output classes. Let $F_{\boldsymbol{\theta}}(\boldsymbol{x}) = \arg\max_k f_{\boldsymbol{\theta}}(\boldsymbol{x})_k$ be the class label predicted by $f_{\boldsymbol{\theta}}$ for any $\boldsymbol{x} \in \mathcal{X}$, where $f_{\boldsymbol{\theta}}(\boldsymbol{x})_k$ is the $k^{\text{th}}$ component of $f_{\boldsymbol{\theta}}(\boldsymbol{x})$.

[1]Department of Information Technology and Electrical Engineering, ETH Zürich [2]Institute for Machine Learning, ETH Zürich. Correspondence to: Rahul Rade <rarade@ethz.ch>.

*Accepted by the ICML 2021 workshop on A Blessing in Disguise: The Prospects and Perils of Adversarial Machine Learning.* Copyright 2021 by the author(s).

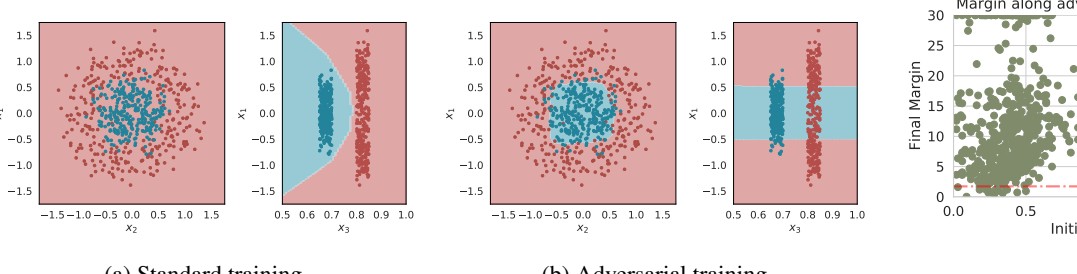

(a) Standard training      (b) Adversarial training

Figure 2. Decision boundary learnt by MLP visualized in two dimensions: $x_1$-$x_2$ and $x_1$-$x_3$ respectively. Adversarial training improves robustness substantially from 41% to 74%, yet causes about 9% drop in accuracy. For each left subplot, $x_3 = 0.85$ and for each subplot on the right, $x_2 = 0.4$.

Figure 3. Final margins for adversarially trained model vs. initial margins before the start of adversarial fine-tuning on CIFAR-10 along $\mathcal{R}_{\text{init}}$. The red dashed line indicates the value of PGD radius $\varepsilon$ used.

**Margin.** Given a classifier $F_{\boldsymbol{\theta}}$, input $\boldsymbol{x}$ and an unit vector $\hat{\boldsymbol{r}} \in \mathbb{S}^{d-1}$, we define margin $\mu$ at $\boldsymbol{x}$ along the direction $\hat{\boldsymbol{r}}$ as:

$$\mu(\boldsymbol{x}, \hat{\boldsymbol{r}}) = \arg\min_{\alpha} |\alpha| \ \text{ s.t. } \ F_{\boldsymbol{\theta}}(\boldsymbol{x} + \alpha\hat{\boldsymbol{r}}) \neq F_{\boldsymbol{\theta}}(\boldsymbol{x}) \quad (1)$$

Additionally, note that we refer to a deep network trained only on clean samples as *standard network* and a network trained via adversarial training as *robust network*. Besides, we reuse the definitions of clean (natural) and robust (adversarial) accuracy as stated by Zhang et al. (2019).

**Initial Adversarial Directions.** Given a standard network $f_{\boldsymbol{\theta}}$, input dataset $\{(\boldsymbol{x}_i, y_i)\}_{i=1}^{n}$, we define the set of initial adversarial directions as $\mathcal{R}_{\text{init}} = \{\boldsymbol{r}_i/||\boldsymbol{r}_i||_2\}_{i=1}^{n}$ where $\boldsymbol{r}_i$ is obtained by solving:

$$\boldsymbol{r}_i = \max_{\boldsymbol{\delta}:||\boldsymbol{\delta}||_p \leq \varepsilon} \ell(y_i, f_{\boldsymbol{\theta}}(\boldsymbol{x}_i + \boldsymbol{\delta})). \quad (2)$$

where, $\ell(\cdot, \cdot)$ is an arbitrary loss function e.g. cross-entropy (CE). This optimization problem is usually solved via projected gradient descent (PGD) (Madry et al., 2018).

## 3. Adversarial Training Leads to Excessive Directional Margin

We begin our analysis by examining the effect of adversarial training (AT) on the decision boundary of DNNs. First, via novel experiments on a toy dataset and CIFAR-10, we show that AT triggers a superfluous increase in the margin along the initial adversarial directions as compared to the nominal increase required to attain robustness. Second, we provide evidence which signifies a direct connection between the increase in margin and reduction in clean accuracy.

**Toy Problem.** First, we study a toy setting to shed some light on the phenomenon of excessive directional margin caused by AT. We construct a 3-d binary classification dataset drawn from two distributions which live on two noisy concentric circles of different radii in the $x_1$-$x_2$ plane

and being linearly separable along the third dimension $x_3$. In particular, $x_1 = \rho_i \cos(z) + \epsilon_1$, $x_2 = \rho_i \sin(z) + \epsilon_2$ and $x_3 \sim \mathcal{U}(\alpha_i, \beta_i)$ where $z \sim \mathcal{U}(0, 2\pi)$ and $\epsilon_1, \epsilon_2 \sim \mathcal{N}(0, \sigma^2)$ where $i = 1, 2$ for class 1 and 2 respectively. We train a single hidden-layer MLP via both standard training and adversarial training. Fig. 2 visualizes the decision regions with both the training procedures. It is evident that the network primarily uses $x_3$ to achieve zero classification error when trained using standard training, but the resulting model performs poorly in terms of robustness. In contrast, when we use AT, the learned decision boundary is completely different from that in the standard case. Here, the network becomes reasonably invariant along $x_3$ (Fig. 1b), thus causing the directional margin along $x_3$ to tend to $\infty$. This enables the network to attain robustness at the cost of a small increase in classification error.

**Evidence on CIFAR-10.** Next, we illustrate that a similar phenomenon occurs in the case of state-of-the-art deep networks trained on CIFAR-10 (Krizhevsky, 2009). To this end, we measure directional margin (as defined by Eq. (1)). We restrict ourselves to the following setting. We take a ResNet-18 (He et al., 2016) trained until convergence on CIFAR-10 (achieving 94.6% accuracy and 0% robustness to $\ell_{\infty}$-PGD on the test set) and then fine-tune it using AT. This framework allows us to study the evolution of the decision boundary caused by AT in comparison to that learnt by a standard network. We use $\ell_{\infty}$-PGD with norm $\varepsilon = 8/255$ for training. The network attains 83.3% accuracy and 51.6% robustness on the test set after adversarial fine-tuning.

During adversarial fine-tuning, we track the margins along the adversarial directions found by PGD to shed some light on the learning dynamics. Suppose $\mathcal{R}_k = \{\boldsymbol{r}_i^k/||\boldsymbol{r}_i^k||_2\}$, where $\boldsymbol{r}_i^k$ denotes the perturbation found by PGD at $k^{\text{th}}$ epoch of fine-tuning for the input sample $\boldsymbol{x}_i$. Thus, $\mathcal{R}_{\text{init}} = \mathcal{R}_0$ represents the set of initial adversarial directions. We hypothesize that during adversarial training, the network

becomes excessively robust to these initial adversarial directions $\mathcal{R}_0 (= \mathcal{R}_{\text{init}})$ while slightly shifting the decision boundaries along other adversarial directions $\mathcal{R}_{k \geq 1}$. Fig. 3 illustrates the margins along $\mathcal{R}_0$ before and after adversarial fine-tuning respectively. The dashed red line indicates the value of $\varepsilon$ used for training, i.e., any allowable perturbation $r$ has $||r||_\infty \leq 8/255$ or equivalently $||r||_2 \leq 1.74$. Intuitively, one might expect adversarial fine-tuning to cause small shifts in the decision boundary so that the margin becomes greater than 1.74, and attain robustness. However, this is not the case in practice. Intriguingly, the classifier instead resorts to becoming largely insensitive along $\mathcal{R}_0$ for some data points as evident in Fig. 3 while undergoing small shifts along other directions $\mathcal{R}_{k \geq 1}$ (see App. C). We also observe a decrease of 11.3% in accuracy after fine-tuning.

**Connection between Margin along $\mathcal{R}_{\text{init}}$ and Clean Accuracy.** We now provide a two-fold argument which justifies the following hypothesis: *The drastic rise in the margin along $\mathcal{R}_{\text{init}}$ is directly correlated to the observed reduction in accuracy. In fact, a larger margin contributes to a larger drop in accuracy.* (i) Firstly, we complement our hypothesis with the following observation by Ortiz-Jimenez et al. (2020). The directions of input space with small margins and in turn, the initial adversarial directions in the case of the standard network, are associated with discriminative features learnt by the network. We believe that these directions are crucial for the performance of the network. Thus, a drastic directional margin along these directions might be the reason for a reduction in clean accuracy. This argument is in line with the previous works by Jetley et al. (2018); Ilyas et al. (2019). (ii) We train a robust network on CIFAR-10 using TRADES (Zhang et al., 2019) with different values for the trade-off parameter $\beta$. As $\beta$ increases, we observe an increase in the margin along $\mathcal{R}_{\text{init}}$ (computed on a subset of 1024 examples from the CIFAR-10 test set) and a corresponding reduction in clean accuracy (see Table 1). This further corroborates our hypothesis. Lastly, we also acknowledge the fact that robust accuracy also improves as the margin increases in the case of TRADES. So, it is not immediately clear if it is possible to reduce the margin while maintaining the same robust accuracy. But, as we will later see with our proposed HAT, this seems to be feasible.

*Table 1.* Median margin along $\mathcal{R}_{\text{init}}$ and the corresponding clean and robust accuracy with TRADES on CIFAR-10 test set for different values of $\beta$. The robust accuracy is evaluated using AutoAttack (Croce & Hein, 2020).

| $\beta$ | Median Margin | Clean | Robust |
|---|---|---|---|
| 1.0 | 8.3 | 88.1 | 43.8 |
| 2.0 | 9.3 | 85.6 | 46.3 |
| 3.0 | 9.7 | 84.7 | 47.9 |
| 4.0 | 10.3 | 83.6 | 48.5 |
| 5.0 | 10.5 | 82.9 | 48.8 |

## 4. Helper-based Adversarial Training

As demonstrated in Sec. 3, adversarial training triggers an unwarranted increase in the margin along initial adversarial directions, thus hindering the network from using highly discriminative features in those directions. In this section, we introduce our proposed algorithm, *Helper-based Adversarial Training (HAT)*, to reduce the excessive directional margin. To this end, we incorporate additional training examples, called helper examples, which are generated on-the-fly during the adversarial training procedure. In particular, a helper example is constructed by extrapolating the adversarial perturbation found during training and is (possibly wrongly) labelled by a normally trained network (see Fig. 4). Formally, we define a helper example as follows.

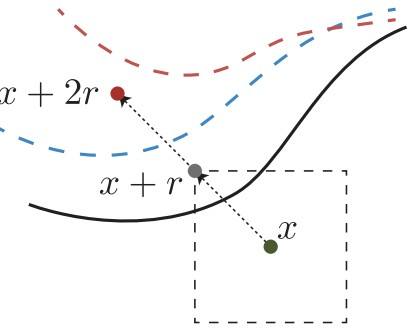

*Figure 4.* Illustration depicting the purpose behind introducing helper examples with labels computed by a standard classifier. Solid black line: standard network. Dashed red line: adversarially trained network. Dashed blue line: desired decision boundary.

**Helper Examples.** Given an input sample $(\boldsymbol{x}_i, y_i)$, a standard network $f_{\boldsymbol{\theta}_{\text{std}}}$, a robust network iterate $f_{\boldsymbol{\theta}_{\text{rob}}^k}$ at $k^{\text{th}}$ training iteration and adversarial example $\boldsymbol{x}_i'$ computed by the adversary $\varphi$ for $f_{\boldsymbol{\theta}_{\text{rob}}^k}$, the corresponding helper example is given by $(\tilde{\boldsymbol{x}}_i, \tilde{y}_i)$ where

$$\tilde{\boldsymbol{x}}_i = \boldsymbol{x}_i + 2\,(\boldsymbol{x}_i' - \boldsymbol{x}_i) \text{ and } \tilde{y}_i = \arg\max_k f_{\boldsymbol{\theta}_{\text{std}}}(\boldsymbol{x}_i')_k$$

The motivation behind this definition is illustrated in Fig. 4 where it is evident that we can stimulate a slight push to the decision boundary along any adversarial direction by making the network predict the correct label $y_i$ at adversarial example $\boldsymbol{x}_i'$ and have it predict $\tilde{y}_i$ (often $\tilde{y}_i \neq y_i$) at helper example $\tilde{\boldsymbol{x}}_i$ to relatively preserve the discriminative characteristics as modelled by a standard network. In contrast to adversarial training, this allows preventing the undesirable excessive rise in the margin to some extent, thus making it possible to achieve significantly better performance on clean samples. Put differently, HAT can also be framed as performing student-teacher learning particularly, geometric self-distillation (Hinton et al., 2015) to mimic certain geometric properties of a standard trained network. We choose to instantiate HAT algorithm by extending TRADES (Zhang

*Table 2.* Comparison of HAT with other adversarial defenses under $\ell_\infty$ adversary ($\varepsilon = 8/255$). We report the mean scores over 3 runs.

| Algorithm | CIFAR-10 | | CIFAR-100 | | SVHN | | TI-200 | |
|---|---|---|---|---|---|---|---|---|
| | Clean | Robust | Clean | Robust | Clean | Robust | Clean | Robust |
| Standard | 94.57 | 0.0 | 76.00 | 0.0 | 96.14 | 0.14 | 65.02 | 0.0 |
| AT | 84.01 | 47.74 | 57.50 | 23.88 | 92.57 | 46.33 | 47.76 | 17.92 |
| TRADES | 82.73 | 48.80 | 57.26 | 23.54 | 91.01 | 52.99 | 48.25 | 17.17 |
| MART | 79.52 | 47.98 | 50.82 | 24.52 | 91.30 | 48.46 | - | - |
| HAT | 84.90 | 49.08 | 59.19 | 23.75 | 93.08 | 52.83 | 52.60 | 18.14 |

et al., 2019) which allows us to balance the accuracy vs. robustness trade-off. Thus, in comparison to TRADES, we have three different loss terms: (i) CE loss on clean samples, (ii) KL divergence loss on adversarial samples weighted by scalar $\beta$, (iii) an additional CE loss on helper samples weighted by $\gamma$. See App. D for the detailed pseudocode of HAT. Note that our extension can be easily incorporated into other recent robust optimization algorithms (Wang et al., 2020; Wu et al., 2020; Zhang et al., 2020; 2021).

## 5. Experiments

In this section, we empirically evaluate the performance of HAT. We report results using ResNet-18 (He et al., 2016) on four datasets: CIFAR-10, CIFAR-100 (Krizhevsky, 2009), SVHN (Netzer et al., 2011) and Tiny-ImageNet-200 (TI-200) (Deng et al., 2009). We compare HAT with three popular adversarial defenses: (i) AT (Madry et al., 2018), (ii) TRADES (Zhang et al., 2019) and (iii) MART (Wang et al., 2020). Please refer to App. E.1 for training details. To evaluate the robustness of our models, we use AutoAttack (Croce & Hein, 2020) which comprises of an ensemble of four diverse attacks for a reliable evaluation of robustness.

Table 2 reports the performance of HAT and other prominent defenses in the literature. It is evident that HAT can significantly improve the clean accuracy of the models while not compromising for robustness. In other words, HAT consistently lowers the gap between accuracy and robustness by $\sim 2\text{-}4\%$ compared to existing adversarial training schemes. For example, in the case of CIFAR-10, HAT provides a $2.2\%$ improvement in clean accuracy whilst achieving a similar robust accuracy as TRADES; for TI-200, we see $4.8\%$ gain in accuracy in comparison to AT. Besides, we also compare the accuracy vs. robustness trade-off obtained by AT, TRADES and HAT on CIFAR-10 in Fig. 5. HAT, by incorporating additional helper samples, clearly outperforms AT and TRADES in terms of the trade-off curve. In addition, we find that the $2.2\%$ gain in clean accuracy on CIFAR-10 also equates to a $2\%$ improvement on common corruptions (Hendrycks & Dietterich, 2019) (see App. E.3).

Finally, we remark that the improvement in the performance on clean samples provided by HAT can be accredited to the

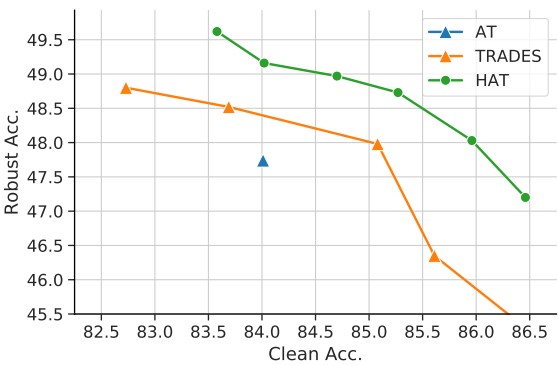

*Figure 5.* Accuracy vs. robustness trade-off exhibited by different adversarial defenses. From left to right, we decrease the trade-off parameter $\beta$ for TRADES and HAT ($\gamma$ is fixed to 0.25).

following two observations on CIFAR-10: (i) HAT exhibits a slightly lower directional margin along $\mathcal{R}_{\text{init}}$ compared to AT and TRADES. (ii) HAT marginally compromises robustness to $\ell_\infty$ perturbations with a larger norm ($\varepsilon > 8/255$). See App. E.4 for more details.

## 6. Conclusion

We presented experimental evidence to highlight that state-of-the-art adversarial defenses foster a superfluous increase in the margin along certain adversarial directions of the input space. This largely destroys the discriminative characteristics along these directions and partly contributes to the much-debated accuracy vs. robustness trade-off. Further, inspired by our analysis, we introduced a novel algorithm, *Helper-based Adversarial Training (HAT)*, to alleviate the problem of excessive directional margin. HAT attempts to mimic the discriminative features learnt by standard trained networks to improve the accuracy on clean samples, hence achieving a superior accuracy vs. robustness trade-off compared to existing defenses. Finally, we verify that HAT slightly reduces the directional margin, thus directly benefiting the accuracy. We believe that our experiments and the proposed HAT algorithm can open the door for further research on comprehending adversarial examples and thus, ameliorating the accuracy vs. robustness trade-off.

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

## A. Toy Problem

Fig. 6 illustrates the toy dataset used for our experiment from two different viewpoints. To be precise, we draw the 3 features $x_1, x_2$ and $x_3$ as follows. $x_1 = \rho_i \cos(z) + \epsilon_1$, $x_2 = \rho_i \sin(z) + \epsilon_2$ and $x_3 \sim \mathcal{U}(\alpha_i, \beta_i)$ where $z \sim \mathcal{U}(0, 2\pi)$ and $\epsilon_1, \epsilon_2 \sim \mathcal{N}(0, \sigma^2)$ where $i = 1, 2$ for class 1 and 2 respectively. We set $\sigma = 0.2$, $\rho_1 = 0.35$, $\rho_2 = 1$, $\alpha_1 = 0.65$, $\beta_1 = 0.70$, $\alpha_2 = 0.80$ and $\beta_2 = 0.85$.

We use a single hidden layer MLP with 25 hidden units and ReLU activation. For training, we use SGD with momentum 0.9, weight decay 0.0005 and set the learning rate to 0.1. We train the model via standard training and adversarial training respectively for 100 epochs. We use $\ell_\infty$ PGD with step size $\alpha = 0.05$, maximum perturbation radius $\varepsilon = 0.1$ and run $K = 5$ iterations for AT.

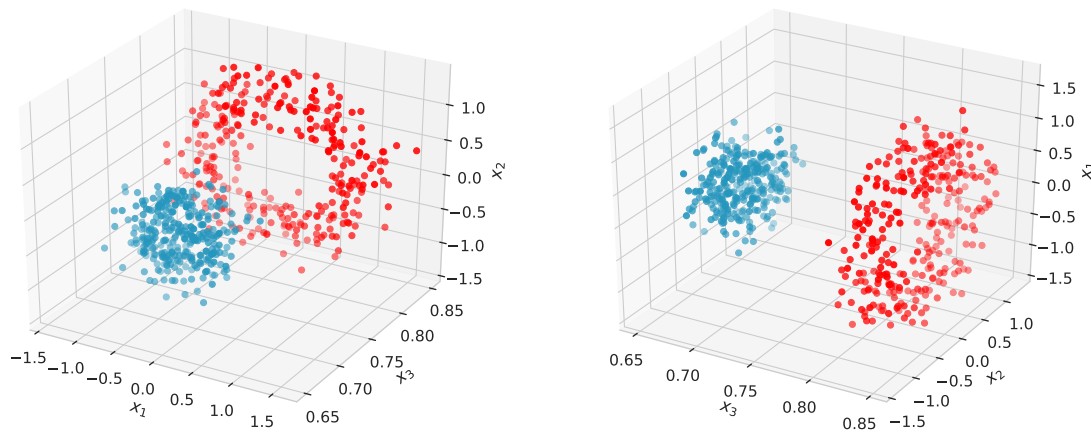

*Figure 6.* Toy dataset used in our experiment.

## B. Experimental Setup For Section 3: Evidence on CIFAR-10

For this experiment, we train ResNet-18 (He et al., 2016) on CIFAR10 (Krizhevsky, 2009) training set. We simply adopt the set of hyperparameters and some improvements from DAWNBench (Coleman et al., 2017) submissions. We use SGD optimizer with Nesterov momentum 0.9 (Nesterov, 1983) and weight decay 0.0005. We further use cyclic learning rates (Smith & Topin, 2018) with cosine annealing and a maximum learning rate of 0.21. We train the model for 50 epochs via standard training. Then, we perform adversarial fine-tuning for 25 epochs with the same scheduler and learning rate settings. Further, we evaluate margins on a set of 512 samples drawn uniformly at random from CIFAR10 test set for the visualizations in Fig. 3 (and Fig. 7).

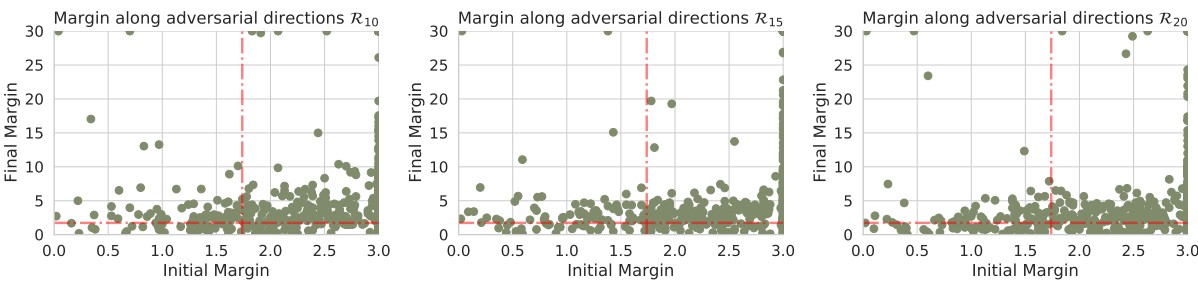

*Figure 7.* Final margins for adversarially trained model vs. initial margins before the start of adversarial fine-tuning on CIFAR-10 along $\mathcal{R}_{10}$, $\mathcal{R}_{15}$ and $\mathcal{R}_{20}$ respectively. The red dashed line indicates the value of $\varepsilon$ used during training and evaluation.

## C. Additional Margin Plots For Section 3: Evidence on CIFAR-10

The margins along other adversarial directions $\mathcal{R}_{10}, \mathcal{R}_{15}, \mathcal{R}_{20}$ before and after adversarial fine-tuning are displayed in Fig. 7. Here, $\mathcal{R}_k$ corresponds to the adversarial directions computed at the $k^{\text{th}}$ iteration of adversarial fine-tuning. We see an increase in the margin along other adversarial directions as expected. Nevertheless, the relative increase along these directions is not as large as compared to that along initial ones $\mathcal{R}_0$ (see Fig. 3).

## D. HAT Algorithm

The pseudocode for helper-based adversarial training with $\ell_\infty$-PGD perturbations is presented in Algorithm 1.

---

**Algorithm 1** Helper-based Adversarial Training

---

    **Input:** Training dataset $\mathcal{D} = \{(\boldsymbol{x}_i, y_i)\}_{i=1}^n$
    **Parameter:** Batch size $m$, learning rate $\eta$, weight of robust loss $\beta$, weight of helper loss $\gamma$, attack radius $\varepsilon$, attack step size $\alpha$ and number of attack iterations $K$

1:  Train a network $f_{\boldsymbol{\theta}_{\text{std}}}$ via standard training on $\mathcal{D}$ i.e., $\boldsymbol{\theta}_{\text{std}} \leftarrow \arg\min_{\boldsymbol{\theta}} \sum_{i=1}^n \text{CE}(y_i, f_{\boldsymbol{\theta}}(\boldsymbol{x}_i))$
2:  Randomly initialize the network parameters $\boldsymbol{\theta}_{\text{HAT}}$
3:  **repeat**                                                                 ▷ Train a robust classifier
4:     Sample a mini-batch $\{(\boldsymbol{x}_{i_j}, y_{i_j})\}_{j=1}^m$ from $\mathcal{D}$
5:     **for** $j = 1, 2, ..., m$ **do**
6:         $\boldsymbol{x}'_{i_j} \leftarrow \boldsymbol{x}_{i_j} + 0.001 \cdot \mathcal{N}(0, I)$                                  ▷ Construct adversarial example
7:         **for** $k = 1, 2, ..., K$ **do**
8:             $\boldsymbol{x}'_{i_j} \leftarrow \prod_{\mathcal{B}(\boldsymbol{x}_{i_j}, \varepsilon)}(\boldsymbol{x}'_{i_j} + \alpha \cdot \text{sign}(\nabla_{\boldsymbol{x}'_{i_j}} \text{KL}(f_{\boldsymbol{\theta}_{\text{HAT}}}(\boldsymbol{x}_{i_j}), f_{\boldsymbol{\theta}_{\text{HAT}}}(\boldsymbol{x}'_{i_j}))))$
9:         **end for**
10:       Compute helper example: $\tilde{\boldsymbol{x}}_{i_j} \leftarrow \boldsymbol{x}_{i_j} + 2\,(\boldsymbol{x}'_{i_j} - \boldsymbol{x}_{i_j})$
11:       Set helper label: $\tilde{y}_{i_j} \leftarrow \arg\max_k f_{\boldsymbol{\theta}_{\text{std}}}(\boldsymbol{x}'_{i_j})_k$
12:     **end for**
13:     $\boldsymbol{\theta}_{\text{HAT}} \leftarrow \boldsymbol{\theta}_{\text{HAT}} - \frac{\eta}{m} \cdot \sum_{j=1}^m \nabla_{\boldsymbol{\theta}_{\text{HAT}}} \Big(\text{CE}\left(y_i, f_{\boldsymbol{\theta}_{\text{HAT}}}(\boldsymbol{x}_{i_j})\right) + \beta \cdot \text{KL}(f_{\boldsymbol{\theta}_{\text{HAT}}}(\boldsymbol{x}_{i_j}), f_{\boldsymbol{\theta}_{\text{HAT}}}(\boldsymbol{x}'_{i_j})) + \gamma \cdot \text{CE}(\tilde{y}_{i_j}, f_{\boldsymbol{\theta}_{\text{HAT}}}(\tilde{\boldsymbol{x}}_{i_j}))\Big)$
14: **until** training completed

---

## E. Further Performance Evaluation

### E.1. Detailed Experimental Setup For Section 5

In this section, we list all the details of our training and evaluation setup. We run our experiments on NVIDIA GeForce GTX 1080 Ti GPUs and use `PyTorch 1.6.0`. We run all our experiments thrice and report the average scores obtained unless stated otherwise. We omit the standard deviations while noting that they are usually small.

**Training Setup.** We use ResNet-18 (He et al., 2016) for CIFAR-10 and CIFAR-100 (Krizhevsky, 2009); and PreAct ResNet-18 for SVHN (Netzer et al., 2011) and Tiny-Imagenet-200 (TI-200) (Deng et al., 2009). For all our experiments, we use SGD optimizer with Nesterov momentum 0.9 (Nesterov, 1983) and weight decay 0.0005. We further employ cyclic learning rates (Smith & Topin, 2018) with cosine annealing and a maximum learning rate of 0.21 for CIFAR-10 and

*Table 3.* Hyperparameters of TRADES, MART and HAT used for training the models reported in Table 2. Here, $\beta$ is the weight of robust loss in TRADES, MART and HAT objective; $\gamma$ is the weight of helper loss in HAT objective.

| Algorithm | CIFAR-10 | CIFAR-100 | SVHN | TI-200 |
|---|---|---|---|---|
| TRADES | $\beta = 5.0$ | $\beta = 5.0$ | $\beta = 5.0$ | $\beta = 8.0$ |
| MART | $\beta = 5.0$ | $\beta = 5.0$ | $\beta = 5.0$ | - |
| HAT | $\beta = 2.5, \gamma = 0.5$ | $\beta = 3.5, \gamma = 0.5$ | $\beta = 2.5, \gamma = 0.5$ | $\beta = 1.75, \gamma = 1.0$[1] |

---

[1] We use CE loss for computing adversarial examples and the robust loss during HAT training instead of KL-divergence since we found CE works much better on TI-200. This also explains the poor robustness of TRADES compared to AT due to the use of KL term.

CIFAR-100, TI-200 ; 0.05 for SVHN. For CIFAR-10 and CIFAR-100, we train the models for 50 epochs with a batch size of 128. In the case of SVHN, we only train for 15 epochs; for TI-200, we train for 30 epochs.

For computing adversarial examples during training, we apply $\ell_\infty$-PGD with the following hyperparameters: $\ell_\infty$ norm $\varepsilon = 8/255$, step size $\alpha = 2/255$ and run the attack for $K = 10$ iterations. Note that we re-implement AT, TRADES and MART and train existing methods and HAT according to the aforementioned settings. The hyperparameters of TRADES, MART and HAT used for training the models in Table 2 are summarized in Table 3. We also use the same setup to train a standard model for computing helper labels during HAT training.

**Evaluation Protocol.** During training, we perform early stopping (Rice et al., 2020) i.e., we track the robustness of the model to PGD ($K = 20$) on the test set and select the model that performs the best for further evaluation. In order to benchmark the $\ell_\infty$ robustness, we always test our models against AutoAttack (AA) (Croce & Hein, 2020) using the default code available at `https://github.com/fra31/auto-attack`. AutoAttack comprises an ensemble of four sophisticated attacks (including a black box attack) for a reliable evaluation of robustness.

### E.2. Achieving State-of-the-art Performance with ResNet-18 on CIFAR-10

In this section, we leverage additional pseudo-labelled data in a bid to further reduce the accuracy-robustness gap with ResNet-18 on CIFAR-10 under $\ell_\infty$ perturbations of size $\varepsilon = 8/255$. We train a ResNet-18 model on CIFAR-10 with additional 500k images taken from Tiny Images dataset (Torralba et al., 2008) as provided by Carmon et al. (2019)). For training, we follow the same setup as Carmon et al. (2019) and use early-stopping with PGD$^{20}$ on the test set. We also retrain a ResNet-18 with RST (Carmon et al., 2019) under the same setting. We use $\beta = 6.0$ for RST and $\beta = 3.5$, $\gamma = 0.5$ for HAT. Table 4 presents the evaluation results. Clearly, HAT surpasses Carmon et al. (2019) by 2.64% in accuracy and 0.52% in robustness. Moreover, HAT achieves a superior accuracy ($\uparrow 3.28\%$) compared to the best performing ResNet-18

*Table 4.* Benchmarking state-of-the-art performance with ResNet-18 on CIFAR-10 under $\ell_\infty$ perturbations ($\varepsilon = 8/255$). We report the results of a single run. *Trained by us using the setup as Carmon et al. (2019). §Best performing ResNet-18 model available on RobustBench (Croce et al., 2020). Robust accuracy is evaluated using AutoAttack.

| Algorithm | Clean | Robust |
|---|---|---|
| Carmon et al. (2019)* | 85.02 | 53.92 |
| Sehwag et al. (2021)§ | 84.38 | 54.43 |
| HAT | 87.66 | 54.46 |

model (Sehwag et al., 2021) available on RobustBench (Croce et al., 2020). Note that the best performing ResNet-18 model uses additional $\sim$ 6M synthetically generated images during training.

### E.3. Accuracy vs. Robustness Trade-off

**Sensitivity of HAT to $\beta$ and $\gamma$.** We examine the effect of the weight of robust loss $\beta$ and the weight of helper loss $\gamma$ on the accuracy vs. robustness trade-off exhibited by HAT. We show the corresponding trade-off curves for HAT with two different values for $\gamma \in \{0.25, 0.50\}$ in Fig 8. For each curve, we decrease $\beta$ from 4.0 to 1.5 from left to right. As evident, $\beta$ balances the trade-off between accuracy and robustness while $\gamma$ has a negligible effect on the resulting trade-off achieved by HAT.

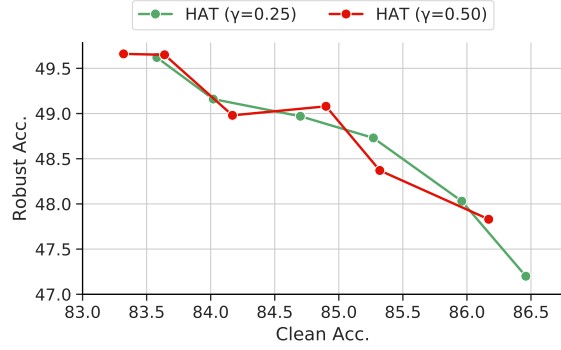

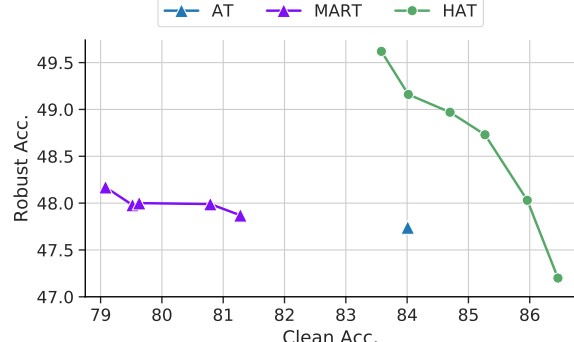

*Figure 8.* HAT accuracy vs. robustness trade-off obtained for different values of $\gamma$. From left to right, we decrease the trade-off parameter $\beta$ for HAT. Robust accuracy is evaluated using AutoAttack.

*Figure 9.* Accuracy vs. robustness trade-off exhibited by AT, MART and HAT. From left to right, we decrease the trade-off parameter $\beta$ for MART and HAT ($\gamma$ is fixed to 0.25). Robust accuracy is evaluated using AutoAttack.

**Comparison with MART.** Fig. 9 compares the trade-off obtained by HAT with that of AT and MART (Wang et al., 2020). For HAT, we fix $\gamma = 0.25$ and vary $\beta \in \{1.5, 2.0, 2.5, 3.0, 3.5, 4.0\}$; for MART, we vary $\beta \in \{1.0, 2.0, 3.0, 4.0, 5.0\}$. HAT surpasses AT and MART by a large margin.

**Evaluation on CIFAR-10 Common Corruptions.** In this part, we examine the performance of HAT on CIFAR-10 common corruptions (Hendrycks & Dietterich, 2019) and compare it with that of AT, TRADES and MART. We compute the accuracy on common corruptions using the default code available at https://github.com/RobustBench/robustbench and evaluate the robust models trained on CIFAR-10 with $\ell_\infty$ perturbations of size $\varepsilon = 8/255$. As seen from the results in Table 5, the $\sim 2\%$ gain in clean accuracy obtained by HAT also leads to a similar gain ($\uparrow 2\%$) in the accuracy under common corruptions. This

*Table 5.* Performance of AT, TRADES, MART and HAT with ResNet-18 on CIFAR-10 common corruptions. We list the mean scores over 3 runs. The robust accuracy is evaluated against AutoAttack.

| Algorithm | Clean | Common Corruptions | Robust |
|---|---|---|---|
| Standard | 94.57 | 72.92 | 0.0 |
| AT | 84.01 | 75.53 | 47.74 |
| TRADES | 82.73 | 74.66 | 48.80 |
| MART | 79.52 | 71.81 | 47.98 |
| HAT | 84.90 | 76.74 | 49.08 |

demonstrates the generality of HAT, also stressing the need to improve both accuracy as well as robustness in order to advance the performance under real-world perturbations.

### E.4. HAT reduces Margin along $\mathcal{R}_{\text{init}}$

We now verify our claim that HAT, to some extent, reduces the unwarranted increase in margin introduced by existing adversarial defenses. To this end, we perform two experiments. Firstly, we take a robust network trained on CIFAR-10 and evaluate it using PGD attack ($K = 40$) for different values of $\ell_\infty$-norm $\varepsilon \in [0, 20/255]$. Fig. 10 plots the difference between robustness of HAT and TRADES vs. $\varepsilon$. We observe that while HAT outperforms TRADES at smaller $\varepsilon$'s, it performs slightly worse after the $\varepsilon$ exceeds the value used during training, i.e., $\varepsilon > 8/255$. This implies that we have traded robustness to high $\varepsilon$'s for an improvement in clean accuracy. Secondly, we take a robust ResNet-18 model trained on CIFAR-10 and evaluate the margin distribution on a random subset of 1024 samples from CIFAR-10 test set along the initial adversarial directions (denoted $\mathcal{R}_{\text{init}}$). The margin for different algorithms illustrated in Table 6. Note that in Table 6, we list the models that achieve the same robust accuracy ($\sim 47.9\%$ to AA). Yet, the median margin for HAT is slightly lower than that for AT and TRADES. This highlights that the decision boundary lies closer to the data for HAT along $\mathcal{R}_{\text{init}}$. Furthermore, this signifies that the predictive power along $\mathcal{R}_{\text{init}}$ is relatively preserved and thus vindicates our claim that HAT benefits clean accuracy (+ 2%) by slightly reducing the directional margin.

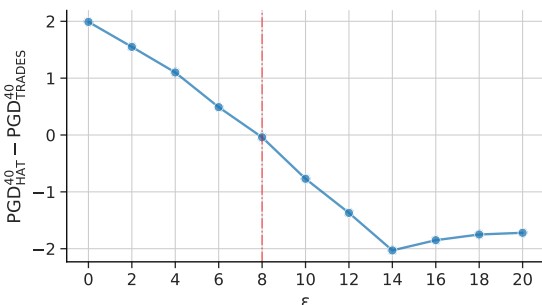

*Figure 10.* Difference between robust accuracy ($\text{PGD}^{40}$) of HAT and TRADES vs $\varepsilon$ ($\ell_\infty$-norm of PGD) on CIFAR-10. The red dashed line corresponds to the value of $\varepsilon$ the models are trained with.

*Table 6.* Median margin for different models along $\mathcal{R}_{\text{init}}$ as computed on a random subset of 1024 samples from CIFAR-10 test set. All the models have $\sim 47.9\%$ robustness to AutoAttack and HAT has the highest clean accuracy. HAT exhibits slightly lower margin along $\mathcal{R}_{\text{init}}$ than AT and TRADES.

| Algorithm | Median Margin |
|---|---|
| AT | 9.3 |
| TRADES | 9.7 |
| HAT | 9.1 |