# OpenReview forum: "Helper-based Adversarial Training: Reducing Excessive Margin to Achieve a Better Accuracy vs. Robustness Trade-off"
_ICML.cc/2021/Workshop/AML — ICML 2021 Workshop AML Oral_

### Official Review · Reviewer_EVjZ · 2021-06-20
**Reducing Excessive Margin to Achieve a Better Accuracy vs. Robustness Trade-off**

**Rating:** Accept
**Confidence:** 5

**Review:**

This paper studied the problem of a trade-off between accuracy and robustness in existing defenses, and present a Helper-based Adversarial Training (HAT), to reduce this effect by incorporating additional wrongly labeled examples during training. In general, the paper is clearly written and easy to follow.  The authors are also encouraged to discuss other backbones to verify the consistent performance, e.g., wide-resnet family.

---

### Decision · Program_Chairs · 2021-06-21

**Decision:**

Accept (Oral)

**Comment:**

A good work studying the trade-off between accuracy and robustness. The paper could provide insights for future research.